# ZERO-SHOT VIDEO SEMANTIC SEGMENTATION BASED ON PRE-TRAINED DIFFUSION MODELS

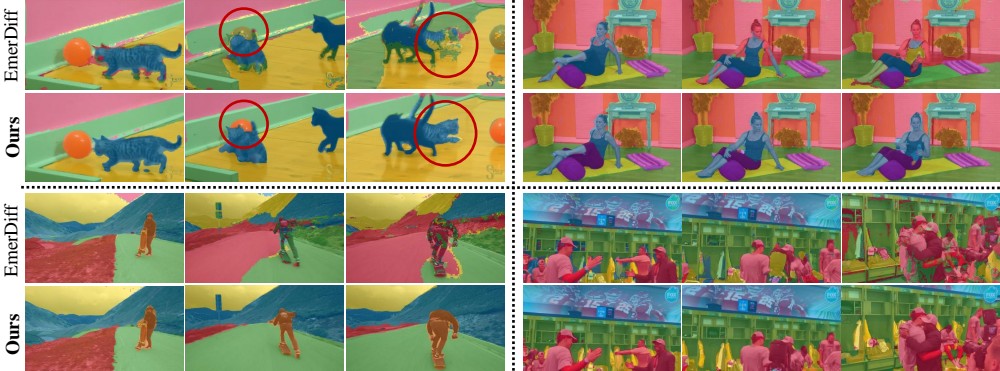

Figure 1: We propose the first *zero-shot* diffusion-based approach for Video Semantic Segmentation (VSS). Our approach produces temporally consistent predictions compared to the diffusion-based image segmentation method EmerDiff (Namekata et al., 2024).

## ABSTRACT

We introduce the first *zero-shot* approach for Video Semantic Segmentation (VSS) based on pre-trained diffusion models. A growing research direction attempts to employ diffusion models to perform downstream vision tasks by exploiting their deep understanding of image semantics. Yet, the majority of these approaches have focused on image-related tasks like semantic segmentation, with less emphasis on video tasks such as VSS. Ideally, diffusion-based image semantic segmentation approaches can be applied to videos in a frame-by-frame manner. However, we find their performance on videos to be subpar due to the absence of any modeling of temporal information inherent in the video data. To this end, we tackle this problem and introduce a framework tailored for VSS based on pre-trained image and video diffusion models. We propose building a scene context model based on the diffusion features, where the model is autoregressively updated to adapt to scene changes. This context model predicts per-frame coarse segmentation maps that are temporally consistent. To refine these maps further, we propose a correspondence-based refinement strategy that aggregates predictions temporally, resulting in more confident predictions. Finally, we introduce a masked modulation approach to upsample the coarse maps to a high-quality full resolution. Experiments show that our proposed approach significantly outperforms existing zero-shot image semantic segmentation approaches on various VSS benchmarks without any training or fine-tuning. Moreover, it rivals supervised VSS approaches on the VSPW dataset despite not being explicitly trained for VSS.

## 1 INTRODUCTION

Diffusion models (Ho et al., 2020; Rombach et al., 2022; Saharia et al., 2022; Ramesh et al., 2022; Blattmann et al., 2023) have showcased remarkable capabilities in learning complex data distributions effectively. This was achieved by exploiting their scalability to train on large-scale datasets (Bain et al., 2021; Schuhmann et al., 2022), allowing them to generate high-quality images and videos with

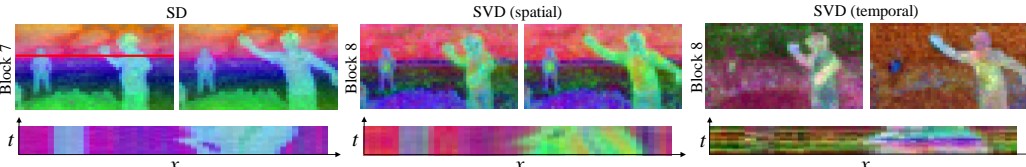

Figure 2: A visualization of the first three PCA components for the features of two video frames extracted from the most semantically rich blocks in SD (Block 7) and SVD (Bock 8). In the second row, we show the $x$-$t$ slice of an image row (highlighted in the red line in the leftmost image) horizontally across the PCA visualization ($x$-axis) and stack it chronologically across the full batch of video frames ($t$-axis). The plot shows that the spatial features of both SD and SVD are temporally more consistent between video frames compared to the features of temporal layers in SVD.

soaring diversity and fidelity. Interestingly, those models could learn a profound understanding of images and their semantics as an indirect consequence of their large-scale training. This entitled them to be considered as *Foundation Models* with high degrees of generalizability and comprehension of images. As a result, a growing research direction attempts to use the internal representations of diffusion models to perform various downstream image vision tasks. For instance, pre-trained diffusion models were used to perform *image vision tasks* such as semantic correspondence (Tang et al., 2023; Luo et al., 2023b), keypoints detection (Hedlin et al., 2023), and semantic segmentation (Namekata et al., 2024; Marcos-Manchón et al., 2024).

Video vision tasks, on the other hand, have not received the same attention compared to their image counterparts. A simple approach is adopting image-based approaches to solve video tasks. To investigate this, we test the diffusion-based image semantic segmentation approach EmerDiff (Namekata et al., 2024) on the task Video Semantic Segmentation (VSS) in a frame-by-frame manner. VSS aims to predict a semantic class for every pixel in each frame according to the pre-defined categories in a video. Our initial experiments show that EmerDiff performs poorly in segmenting videos in terms of temporal consistency, as shown in Figure 1. This can be attributed to the lack of modeling of video temporal information, causing inconsistent predictions across frames.

An intuitive solution for enhancing the temporal consistency of these approaches is employing a video diffusion model, *e.g.*, Stable Video Diffusion (SVD) (Blattmann et al., 2023) as a backbone. The SVD model is initially trained on images, then expanded with temporal layers and fine-tuned on videos. Ideally, the temporal features from SVD should exhibit better temporal consistency. To investigate this hypothesis, we visualize the temporal features of a pre-trained SVD in Figure 2 to examine their temporal consistency. The figure shows that the temporal features are surprisingly unstable and tend to change significantly between video frames. On the other hand, the spatial features encode the structure of the scene, similar to Image Stable Diffusion (Rombach et al., 2021) (SD). Based on these observations, we capitalize on the spatial features of either SD or SVD and attempt to enhance them temporally.

In this paper, we introduce a diffusion-based zero-shot approach for VSS. First, we build a *scene context model* based on the features of a pre-trained image (SD) or video (SVD) diffusion model. This context model predicts per-frame coarse segmentation maps and is autoregressively updated to accommodate scene changes throughout the video. To further enhance the temporal and spatial consistency of these coarse maps, we propose a correspondence-based refinement (CBR) strategy encompassing a pixel-wise voting scheme between the video frames. Finally, we propose a masked modulation process to reconstruct the full-resolution segmentation maps that is more stable and less noisy than that of Namekata et al. (2024). Experiments show that our proposed approach significantly outperforms zero-shot image semantic segmentation methods on various VSS benchmarks. More specifically, we improve mIOU over image semantic approaches by at least 29% on VSPW, CityScapes, and Camvid datasets. Remarkably, our approach performs comparably well as supervised VSS approaches on the diverse VSPW dataset. We also show that for currently released models, SD features lead to a higher quality result than SVD features, but this trend may reverse when SVD training considers larger datasets in the future.

## 2 RELATED WORK

### 2.1 DIFFUSION MODELS' FEATURES

The large-scale training of diffusion models on the LAION-5B dataset (Schuhmann et al., 2022) with 5 billion images allowed them to learn semantically rich image features. As a result, a growing direction of research attempts to employ these features to perform downstream vision tasks. Several approaches (Luo et al., 2023b; Tang et al., 2023; Zhang et al., 2024) investigated using these features to perform semantic correspondence. They observed that the features are semantically meaningful and generalize well across different objects and styles of images. For example, a human head in any arbitrary image will have similar features to any other human head in other images, regardless of the scene variations. Those features even generalize across similar classes of objects like animal heads. At the same time, the features will differ from those of unrelated object classes like buildings, landscapes, vehicles, *etc*. EmerDiff (Namekata et al., 2024) capitalized on this observation to perform image semantic segmentation. Since the diffusion features are distinct for different objects, they can easily be clustered to separate those objects and produce a coarse segmentation map. Then, they proposed a modulation strategy to produce fine segmentation maps at a remarkable quality. However, we observed that the produced segmentation maps by EmerDiff are not temporally consistent, as illustrated in Figure 1, making it unsuitable for Video Semantic Segmentation (VSS). Therefore, we propose a diffusion-based pipeline tailored for VSS with a focus on improving temporal consistency.

### 2.2 VIDEO SEMANTIC SEGMENTATION

Video semantic segmentation (VSS) (Wang et al., 2021; Zhang et al., 2023a;b; Li et al., 2024; Zhao et al., 2017; Cheng et al., 2021; Yang et al., 2022) is a spatiotemporal variation of image segmentation on videos that aims to predict a pixel-wise label across the video frames. Those predictions should be temporally consistent under object deformations and camera motion, making VSS more challenging than its image counterpart. Recent approaches attempted to exploit the temporal correlation between video features to produce temporally consistent predictions. Several approaches (Zhu et al., 2017; Gadde et al., 2017) incorporated optical flow prediction to model motion between frames. Other approaches (Liu et al., 2020) proposed a temporal consistency loss on the per-frame segmentation predictions as an efficient replacement for optical flow. TMANet (Wang et al., 2021) utilized a temporal attention module to capture the relations between the current frame and a memory bank. DVIS (Zhang et al., 2023a) further improved the efficiency by treating VSS as a first-frame-segmentation followed by a tracking problem. Recent work UniVS (Li et al., 2024) proposed a single unified model for all video segmentation tasks by considering the features from previous frames as visual prompts for the consecutive frames. Despite their remarkable performance, these supervised approaches do not generalize well on unseen datasets (Zhang et al., 2023b). Therefore, it is desired to have an approach that generalizes well across datasets. Inspired by the success of EmerDiff (Namekata et al., 2024) on image semantic segmentation, we attempt to exploit the diffusion features to propose the first temporally consistent zero-shot VSS approach.

## 3 PRELIMINARIES

### 3.1 STABLE DIFFUSION ARCHITECTURE

Stable Diffusion (SD) (Rombach et al., 2021; Podell et al., 2023) is one of the prominent latent diffusion models that achieves a good tradeoff between efficiency and quality. It is trained to approximate the image data distribution by adding noise to the latents of data samples until they converge to pure Gaussian isotropic noise. During sampling, it performs a series of Markovian denoising steps starting from pure noise to recover a noise-free latent that is decoded to produce a synthetic image. SD utilizes a UNet architecture to predict either the noise or some other signal at each time step. This UNet encompasses multiple blocks for the encoder and the decoder at different resolutions ranging from $8 \times 8$ to $64 \times 64$, where every block has residual blocks, self-attention, and cross-attention modules. The attention is computed as:

$$f\left(\sigma\left(\frac{QK^T}{\sqrt{d}}\right) \cdot V\right) \tag{1}$$

where $Q, K, V$ are the query, key, and value vectors in the attention layers, $\sigma$ is the Softmax activation, and $f$ denotes a fully connected layer. The query is always computed from the image features, while the key and the value are computed from the image in self-attention and a conditional signal (*e.g.* textual prompt) in cross-attention. Since the semantically rich features are located in the decoder (Tang et al., 2023; Namekata et al., 2024), We only consider the decoder blocks. The decoder has 12 blocks over 4 resolutions, where We refer to the first block as 0, with a resolution of $8 \times 8$, and the last block as 11, with a resolution of $64 \times 64$.

## 3.2 Emerging Image Semantic Segmentation from Diffusion Models

EmerDiff (Namekata et al., 2024) observed that it is possible to extract semantically rich features from some of the UNet blocks and use them to produce coarse semantic segmentation maps. Given an RGB image $X$ with a resolution of $H \times W$, the spatial resolution of the low-dimensional UNet features becomes $H/S_i \times W/S_i$, where $S_i$ is the scale factor of block $i$ determined by both the size of the latent representation and the downsampling factor of that block. By applying K-Means clustering on the low-dimensional feature maps from Block $b_k$ at timestep $t_k$, we obtain a set of binary masks $\mathcal{M} = \{M_1, M_2, ..., M_L\}$, where $M \in \mathbb{R}^{H/S_i \times W/S_i}$, and $L$ is the number of distinct clusters.

A *modulation* strategy is used to obtain fine-grained image-resolution segmentation maps. This is achieved by modulating the attention module at block $b_m$ and denoising timestep $t_m$ for each binary mask $M_l$ based on the following formula:

$$f\left(\sigma\left(\frac{QK^T}{\sqrt{d}}\right) \cdot V \pm \lambda M_l\right), \tag{2}$$

where $\lambda$ controls the degree of modulation. The intuition behind this process is to add or subtract a certain amount of perturbation $\lambda$ on the region specified by mask $M_l$ and then continue the denoising process to reconstruct a modulated image. By applying $+\lambda$ and $-\lambda$, we get two different modulated images denoted as $I_l^+$ and $I_l^-$ respectively. Then, a difference map is computed as $D_l = \|I_l^+ - I_l^-\|^2$, where $D_l \in \mathbb{R}^{H \times W}$. This is repeated for all masks in $\mathcal{M}$ to get a set of difference maps $\mathcal{D} = \{D_1, D_2, ..., D_L\}$. Finally, the full-resolution segmentation map is computed as $Y = \arg\max_l \mathcal{D}$.

## 4 Method

Our method encompasses three main components, as illustrated in Figure 3. First, we construct a *scene context model* to produce coarse segmentation maps based on the diffusion features (Section 4.1). Then, we introduce a *correspondence-based refinement* strategy to curate the coarse segmentation maps (Section 4.2). Finally, we propose a *masked modulation* approach that produces less noisy and more stable full-resolution segmentation maps (4.3). We also provide some details on adapting our approach to employ features from Stable Video Diffusion (SVD) in Section 4.4.

## 4.1 Scene Context Model

Image segmentation algorithms are designed for segmenting individual images and can only process videos in a frame-by-frame manner. This is not ideal for videos, as the per-frame predictions will be completely independent and consequently temporally inconsistent. To address this limitation, we propose to create a scene context model that is initialized at the first frame and then updated throughout the video in an autoregressive manner.

Given a video sequence $\mathcal{X} = \{X^1, X^2, ..., X^N\}$ with $N$ frames, we extract diffusion features $F_i^n$ for all frames in $[1, N]$, where $i$ is the decoder block. Since different blocks have different information, we aggregate features from multiple blocks by averaging to produce an aggregated feature $\widetilde{F}^n$. We found that aggregating blocks 6, 7, and 8 attains the best results (see Section 5.4). Note that these blocks share the same resolution and number of channels. Then, we process the video in batches of length $B$.

For the first batch, we use K-Means to extract the initial coarse segmentation map $M^1$ for the first frame based on the aggregated features $\widetilde{F}^1$. Given the diffusion features $\widetilde{F}^1$ and the coarse map

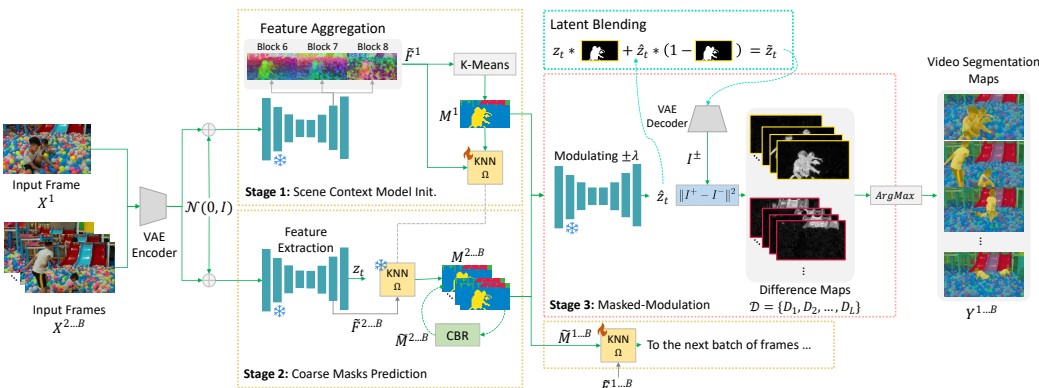

Figure 3: Our Video Semantic Segmentation (VSS) approach encompasses three stages. In Stage 1, we initialize a Scene Context Model $\Omega$ as a KNN classifier with the aggregated diffusion features $\widetilde{F}^1$ of the first frame and a coarse mask $M^1$ produced by K-Means clustering. In Stage 2, we use the context model $\Omega$ to predict coarse masks for the remaining frames in the batch $M^{2\cdots B}$. We refine the coarse maps $M^{1\cdots B}$ using our correspondence-based refinement (CBR). In Stage 3, we use the refined coarse masks to modulate the attention layers of the diffusion process with factor $\pm\lambda$ to obtain a modulated latent $\hat{z}_t$. Then, we blend $\hat{z}_t$ with the original unmodulated latent $z_t$ using the coarse masks to obtain a less noisy latent $\widetilde{z}_t$. Finally, the latent $\widetilde{z}_t$ is decoded to obtain images $I^+, I^-$ that are used to compute a set of difference maps per segment $l \in L$. The final predictions are made by applying an $\arg\max$ operation over the difference maps similar to (Namekata et al., 2024). The process is repeated for the following batch of frames where the context model is updated in an autoregressive manner using the coarse masks $M^{1\cdots B}$ and their corresponding features $\widetilde{F}^{1\cdots B}$.

$M^1$ as labels, we train a KNN classifier $\Omega(M^1, \widetilde{F}^1)$ as a context model to discriminate between different clusters. Then, we use the context model $\Omega$ to predict the coarse segmentation maps for the remaining frames in the first batch $\{M^2, M^3, \ldots, M^B\}$. We refine the coarse maps further using the correspondence-based refinement (Section 4.2) and use them alongside their aggregated diffusion features to update the context model as $\Omega([\widetilde{M}^1, \widetilde{M}^2, \ldots, \widetilde{M}^B], [\widetilde{F}^1, \widetilde{F}^2, \ldots, \widetilde{F}^B])$. The context model is then used for the next batch. This strategy ensures that the context model $\Omega$ adapts to changes in the video in an auto-regressive manner.

## 4.2 CORRESPONDENCE-BASED REFINEMENT

Since the context model operates purely in the feature space, it is unaware of the spatial arrangement of clusters or how they develop temporally. This might cause inconsistencies between clusters, especially across borders between objects. To alleviate this, we propose a refinement strategy based on the semantic correspondence between consecutive frames. We compute per-pixel correspondences (in the coarse map resolution) similar to Tang et al. (2023) based on the diffusion features of block $c$ between images $j$ and $j+1$ to produce a correspondence-based coarse segmentation $\hat{M}^j$. First, we compute a trajectory $T$ for each pixel $p$ in frame $j$ that maps to the most similar pixel $q$ in the following frame $j+1$ as follows:

$$T^j[p] = \arg\max_q \Gamma(F_c^j[p], F_c^{j+1}[q]), \quad \text{with } \| p - q \|^2 \leq \mathcal{T} \tag{3}$$

where $\Gamma$ is a distance metric that we choose to be the cosine similarity. The threshold $\mathcal{T}$ discards faulty matches that are spatially unplausible. These correspondences are computed for all pixels $p$ and over all frames within the batch. Then, we define a recursive tracking function $\texttt{TRACK}$ that follows the trajectory from frame to frame to fetch the corresponding class label:

(a)                                                             (b)

Figure 4: A detailed illustration of (a) the scene context model and (b) correspondence-based refinement.

$$\text{TRACK}(p, j, J) = \begin{cases} \text{TRACK}(T^j[p], j + 1, J), & \text{if } j \leq J \\ p, & \text{if } j > J \end{cases}, \tag{4}$$

Afterward, we employ this function to query class labels for each pixel on the trajectory of pixel $p$. We perform a majority voting for all pixels over the temporal axis to produce the final coarse segmentation map $\widetilde{M}^j$:

$$\widetilde{M}^j[p] = \arg \max_{l \in L} \sum_{k=j}^{B-1} \mathbf{1}\big(M^j[\text{TRACK}(p, j, k)] = l\big) . \tag{5}$$

We compute the counting using the indicator function $\mathbf{1}$, which is equal to one if the condition $M^j[\text{TRACK}(p, j, k)] = l$ is True, and zero otherwise. This interplay between the context model and the correspondence-based refinement leads to more accurate predictions. The context model encodes the feature space of the video batch, while the refinement strategy spatially and temporally regularizes the predictions. This process is illustrated in Figure 4.

## 4.3 MASKED MODULATION

The modulation process aims to upsample the coarse segmentation masks to the full resolution of the video frames. When applying the modulation process, it is only the modulated regions that are expected to change, as explained in Section 3.2. However, in practice, the modulation process produces noise outside that region, causing discrepancies when computing the final segmentation labels. Therefore, we propose a masked modulation process that employs the coarse segmentation map to mask out both the latents and the difference maps outside the modulated region. For a denoising timestep $t$, we blend the latents as follows:

$$\widetilde{z}_t = z_t * (1 - M_l) + \hat{z}_t * M_l, \tag{6}$$

where $z_t$ is the latent from the unmodified sampling step, $\hat{z}_t$ is the latent from the modulated sampling process, and $M_l$ is the low-resolution mask we are modulating. Even though the modulation is only performed at timestep $t_m$, we apply latent blending after $t_m$ until timesteps $t_f$, as once the attention map of a certain timestep is modified, it will influence all the following denoising timesteps.

To further suppress the noise in the difference map, we can apply the same blending strategy to the difference maps. We compute the filtered difference map $\widetilde{D}_l$ as:

$$\widetilde{D}_l = D_l * M_l + s \cdot D_l * (1 - M_l), \tag{7}$$

where $s$ is a scaling hyperparameter that controls the filtering strength.

### 4.4 ADAPTING SVD FOR VIDEO SEMANTIC SEGMENTATION

As it is natural to explore applying video models for video tasks, we further investigate the possibility of using Stable Video Diffusion (SVD)(Blattmann et al., 2023) features to perform VSS. Unlike SD, which is text-conditioned, SVD is image-conditioned. SVD adapts the SD 2.1 architecture and finetunes on a highly-curated large video dataset. Moreover, it employs a video VAE encoder to encode and decode the videos. SVD extends the SD architecture design mainly in two parts: 1) A video residual network that consists of a set of convolutional blocks that handle every frame of the video latent independently. 2) A temporal attention layer is applied on top of the output of the spatial attention layer, which computes the full attention of the features along the temporal axis on each spatial location. The output of the temporal attention and the output of the spatial attention are then mixed with a learnable weight. We use SVD as a backbone model to extract the features as well as do modulation following similar steps described in previous sections.

## 5 EXPERIMENTS

To the best of our knowledge, no zero-shot diffusion-based video semantic segmentation (VSS) approach exists. Therefore, we compare against existing zero-shot image segmentation methods by adapting them to the VSS setting.

### 5.1 EXPERIMENTAL SETUP

**Implementation details.** We evaluate our method with both SD 2.1 and SVD backbones, and we denote them as *Ours (SD)* and *Ours (SVD)*. Given the video frames, we encode them using VAE, and we add a certain level of noise that corresponds to timestep $t_{inv}$ and start the denoising process at this noise level. As we need to perform modulation at timestep $t_m$, the value of $t_{inv}$ must be larger than $t_m$. Initially, we set the modulating coefficient $\lambda$ to 10 as in (Namekata et al., 2024). However, we observed that latent blending suppresses the modulating strength; thus, we increase $\lambda$ to 50 when latent blending is enabled. We set the batch size $B = 14$, which is also the original training batch size of SVD.

We fix all hyperparameters for all videos and datasets, and we do not apply any post-processing methods like a conditional random field (CRF) on the output segmentation maps. All experiments were conducted on a single NVIDIA A100 40G GPU. We provide detailed parameter settings for SD and SVD backbones in the Appendix.

**Evaluation protocol.** The clusters generated by K-Means in the first frame are class-agnostic. To be able to compare our predicted segmentation maps against the groundtruth, we use the labels from the groundtruth of the first frame to assign labels to our clusters. This is similar to the evaluation protocol in Video Object Segmentation, but we differ in that we only use the labels from the groundtruth and keep the segmentation masks from K-Means.

**Baselines.** We compare against zero-shot image segmentation approaches CLIPpy (Ranasinghe et al., 2023) and EmerDiff (Namekata et al., 2024). For EmerDiff, we adapt it to our zero-shot VSS setup. We use the same label assignment strategy for the first frame, but then we train a KNN classifier to predict the next frame in an autoregressive manner, similar to our approach. We also provide results of the supervised approaches DVIS++ (Zhang et al., 2023a), UniVS (Li et al., 2024), and TMANet (Wang et al., 2021) to showcase where our zero-shot approach stands compared to them.

**Dataset.** We evaluate on the validation set of three commonly used VSS datasets: VSPW (Miao et al., 2021) with diverse videos, Cityscapes (Cordts et al., 2016) and CamVid (Brostow et al., 2009) with driving videos. VSPW has a resolution of 480p, while CamVid is 360p. For Cityscapes, to fit the video frames into GPU memory, we generate the segmentation maps at the resolution of $512 \times 256$ and upsample them to $2048 \times 1024$ for evaluation. We provide details of the settings for individual datasets in the Appendix.

**Metrics.** We report mean Intersection-over-Union (mIoU) and mean Video Consistency (mVC) (Miao et al., 2021) as quantitative metrics similar to existing VSS approaches (Zhang et al., 2023a;b; Wang et al., 2021). mIoU describes the mean intersection-over-union between the predicted and ground truth pixels, while mVC computes the mean categories' consistency over the long-range

Table 1: Quantitative performance comparison.

| Method | Backbone | Training | VSPW | | | Cityscapes | Camvid | | |
|---|---|---|---|---|---|---|---|---|---|
| | | | mIoU | mVC$_8$ | mVC$_{16}$ | mIoU | mIoU | mVC$_8$ | mVC$_{16}$ |
| TMANet(Wang et al., 2021) | ResNet-50 | Supervised | – | – | – | 80.3 | 76.5 | – | – |
| UniVS(Li et al., 2024) | Swin-T | Supervised | 59.8 | 92.3 | – | – | – | – | – |
| DVIS++(Zhang et al., 2023a) | VIT-L | Supervised | 63.8 | 95.7 | 95.1 | – | – | – | – |
| CLIPpy | T5 + DINO | zero-shot | 17.7 | 72.4 | 68.4 | 4.7 | 2.6 | 44.4 | 35.6 |
| EmerDiff (SVD) | SVD | zero-shot | 39.7 | 82.1 | 78.5 | 11.0 | 7.3 | 71.7 | 64.4 |
| EmerDiff (SD) | SD 2.1 | zero-shot | 43.4 | 68.9 | 64.3 | 21.5 | 6.9 | 39.8 | 32.9 |
| Ours(SVD) | SVD | zero-shot | 53.2 | 89.3 | 88.0 | 36.2 | 16.6 | 87.4 | 85.8 |
| Ours(SD) | SD 2.1 | zero-shot | **60.6** | **90.7** | **89.6** | **37.3** | **20.6** | **92.3** | **91.9** |

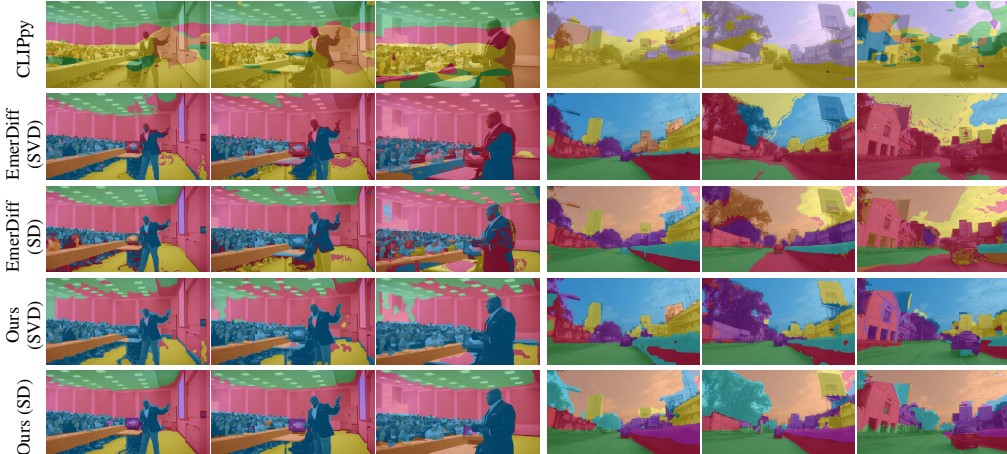

Figure 5: Qualitative comparison of different zero-shot methods. Note that the color of a segmentation cluster only represents the relative index of the clusters when the video is processed. The color itself does not map to an absolute label.

adjacent frames. We denote mVC evaluated under 8 and 16 video frames as mVC$_8$ and mVC$_{16}$, respectively. We use both metrics together to showcase the segmentation quality on individual images as well as the overall temporal consistency.

## 5.2 QUANTITATIVE RESULTS

We provide the quantitative results in Table 1. Our approach with both SD and SVD backbones performs the best in terms of all evaluation metrics on all datasets amongst the zero-shot approaches. More specifically, it improves over EmerDiff for both SD and SVD backbones in terms of mIoU with 33%, 29% on the VSPW dataset, 54%, 106% on the CityScapes dataset, and 99% and 78% on the Camvid dataset. Furthermore, our approach performs similarly to the supervised method UniVS and DVIS++ in terms of mIoU despite not being explicitly trained for this task. On the CityScapes and Camvid datasets, our approach outperforms the zero-shot methods by a huge margin. However, there is still a performance gap compared to the supervised approaches. We attribute this to two main reasons. First, CityScapes and Camvid datasets are for driving scenarios and have small objects and challenging lighting conditions, which poses a challenge when inverting the video frames (see appendix). Secondly, we downsample their video frames of CityScapes by a factor of 16 to match the resolution of the diffusion models. This makes it difficult to segment small objects such as pedestrians and traffic poles, contrary to the VSS-specialized approaches that consider small objects when designing their solutions. We leave it for future work to adopt a tiled approach for high-resolution video segmentation.

Table 2: Ablation study. The videos we use here are the first 30 videos from VSPW validation set. CBR refers to Correspondence-Based refinement.

| Batch size $B$ | Masked Modulation | Feature Aggregation | CBR | SD2.1 | | | SVD | | |
|---|---|---|---|---|---|---|---|---|---|
| | | | | mIoU | mVC$_8$ | mVC$_{16}$ | mIoU | mVC$_8$ | mVC$_{16}$ |
| 1 | ✗ | ✗ | ✗ | 33.4 | 70.6 | 60.2 | 26.6 | 82.2 | 78.3 |
| 14 | ✗ | ✗ | ✗ | 43.4 | 76.9 | 73.8 | 38.9 | 90.2 | 88.7 |
| 14 | ✓ | ✗ | ✗ | 45.6 | 87.5 | 85.5 | 38.6 | 90.5 | 89.0 |
| 14 | ✓ | ✓ | ✗ | 46.6 | 87.6 | 85.7 | **39.4** | 90.2 | 89.2 |
| 14 | ✓ | ✗ | ✓ | **47.4** | 89.2 | 87.5 | 37.1 | 91.5 | 90.3 |
| 14 | ✓ | ✓ | ✓ | 46.5 | **89.8** | **88.4** | 38.2 | **92.3** | **91.3** |

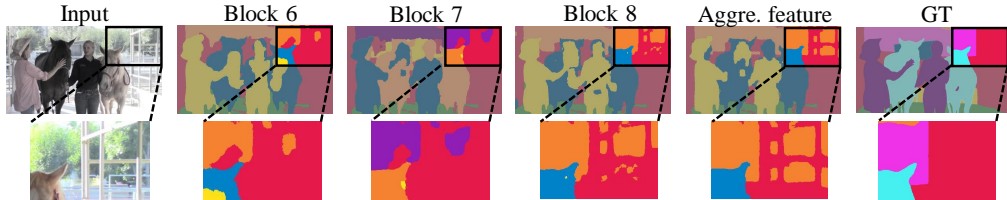

Figure 6: High-resolution segmentation map generated by aggregated feature has more details than features from a single block. We omit the low-resolution segmentation map for a better visual comparison.

## 5.3 QUALITATIVE RESULTS

We show some qualitative results of different zero-shot methods in Figure 5. CLIPpy struggles to locate the boundaries accurately and produce coarse segments. EmerDiff can segment the first frame accurately but struggles to preserve the masks temporally. Our method with SD backbone produces the best segmentation maps in terms of segmentation quality and temporal consistency. The maps have sharper boundaries and clean clusters compared to other methods. Ours (SVD) performs better than its EmerDiff counterpart. However, the overall segmentation quality obtained by the SVD backbone is worse than that of SD. This can be attributed to the degraded feature representation of SVD compared to SD as a result of training on a relatively small video dataset compared to SD.

## 5.4 ABLATION ANALYSIS

We show an ablation of our newly proposed components in Table 2. Incorporating more frames in training our context model greatly improves the mIoU as well as mVC. Enabling the masked modulation and the feature aggregation improves all metrics further. The best results in terms of mIoU are attained by enabling the correspondence-based refinement and disabling the feature aggregation. The feature aggregation can negatively impact the mIoU on some occasions due to the coarse groundtruth of the VSPW dataset. Figure 6 shows an example where our approach remarkably predicts the fine details of the window as one class and the background as another class, while the groundtruth annotates the whole region as a window. Finally, the best trade-off between the segmentation quality (mIoU) and temporal consistency (mVC) is achieved with all components.

**Feature Aggregation.** To validate the efficacy of feature aggregation, we show a visual comparison of the segmentation maps produced by features in different blocks in Figure 6. The aggregated features can encode more spatial details, which could enhance the coarse masks and, consequently, the high-resolution segmentation maps.

**Masked Modulation.** We show the qualitative comparison with or without the latent blending in Figure 7. The figure shows that without the latent blending, the difference map in (c) contains high activations outside of the modulated sub-region indicated by the coarse binary mask. The existence of activation in these regions can lead to a false assignment of the segmentation labels, as shown in (e). For example, a part of the lab table is classified as a wall. After applying latent blending, we remove activations outside of the mask region and obtain a cleaner segmentation mask, as shown in (f).

| Input image | Low-res mask | wo.masked modulat. | w.masked modulat. | wo.masked modulat. | w.masked modulat. |
| --- | --- | --- | --- | --- | --- |
| (a) | (b) | (c) | (d) | (e) | (f) |

Figure 7: We show that given a low-resolution mask of a sub-region (b), the corresponding difference map without masked modulation (c) can have high activation outside the masked sub-region, which will result in spatial inconsistency on the final segmentation map (e). By applying masked modulation on the intermediate latents, the high activation on the irrelevant regions of the difference map will be removed (d), therefore producing a cleaner segmentation map (f).

## 6  LIMITATIONS AND FUTURE WORK

One of the limitations of our approach is its dependency on the quality of the image inversion method. Moreover, fine image details are likely to be discarded due to the compression from the VAE encoder. Therefore, our approach can benefit from future research improving both VAE encoding and image inversion. It is also beneficial to investigate if other video diffusion models have semantically higher quality features than SVD. Another limitation of our approach is that it is instance-agnostic, *i.e.* it groups all objects of the same class into the same cluster. This is due to the inherent nature of diffusion features that group similar semantics. For future work, our approach can be extended to perform Video Instance or Panoptic segmentation.

## 7  CONCLUSIONS

In this work, we introduced the first *zero-shot* method for Video Semantic Segmentation (VSS) using pre-trained diffusion models. We proposed a pipeline tailored for VSS that leverages image and video diffusion features, and attempts to enhance their temporal consistency. Experiments showed that our proposed approach significantly outperforms zero-shot image semantic segmentation methods on several VSS benchmarks, and performs comparably well to supervised methods on VSPW dataset.

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

## A SUPPLEMENTAL MATERIAL

### A.1 CODEBASE AND WEBSITE

Our anonymized code is available at https://anonymous.4open.science/r/VidSeg_Anonymous-827E. We also provide video results on the anonymous project page https://anonymous.4open.science/w/VidSeg_Anonymous-E341-website/.

### A.2 BROADER IMPACT

Video semantic segmentation has important applications in autonomous driving and surveillance security. However, the source videos used for segmentation may contain private information such as human faces and driving plates. Guidelines for responsible usage need to be made to prevent privacy invasion. Also, diffusion models can inherit and propagate biases from their training data, leading to unfair treatment of certain groups.

### A.3 TAXONOMY OF VIDEO SEGMENTATION TASKS

Here we provide a short taxonomy of different video segmentation tasks:

- Video Semantic Segmentation (VSS) aims to predict a semantic class for every pixel according to the pre-defined categories in a video.
- Video Object Segmentation (VOS) aims to segment and track the dominant object(s) in a video.
- Video Instance Segmentation (VIS) aims to segment and track individual instances of object(s) in a video.
- Video Panoptic Segmentation (VPS) aims at segmenting every pixel either into foreground object instances or background semantic classes in a video.
- Promptable Video Segmentation (PVS) it is a new video segmentation paradigm introduced by Ravi et al. (2024) that aims to segment an object through a video as sepcified by a user prompt (point, bounding box, or mask).

### A.4 SETTINGS

**Dataset.** VSPW (Video Scene Parsing in the Wild) is a large-scale video semantic segmentation dataset that consists of a wide range of real-world scenarios and categories. It has 124 categories in total. The resolution of this dataset is $480 \times 853$. Since each video consists of less than 10 classes, we set the number of clusters for KMeans as 20. Cityscapes is a large-scale urban streets video sequence dataset. The objects are grouped into 30 classes in total. Each video clip has 30 frames, and only the 20th frame has dense annotations. We use its validation set, which contains 15000 frames from three cities. As the original resolution of the frames is $1024 \times 2048$, which is too big to fit into the GPU memory with SVD, we downsample the frames to $256 \times 512$ for all the experiments and evaluate at the original resolution by upsampling the segmentation maps. As each video clip may contain classes of more than 10, we set the number of clusters as 30 in order to capture the small objects. CamVid (Cambridge-driving Labeled Video Database) is a road scene dataset with dense segmentation annotations with 11 classes in total. The resolution of this dataset is $360 \times 480$. We use its validation set, which has one video clip and contains 100 frames. We set the number of clusters for KMeans as 20.

**Implementation details.** We build our method on top of SD 2.1 and SVD code repository https://github.com/Stability-AI/generative-models.

**Computational resources.** We use one NVIDIA A100 GPU to conduct all our experiments. As the running time depends on the number of clusters and spatial resolution of the video frames, generally, it will take around 2 minutes to process a batch of video frames using SD 2.1 and 5 minutes using SVD. The main cost of the time comes from the modulating process, which involves several modified forward passes of the backbone model.

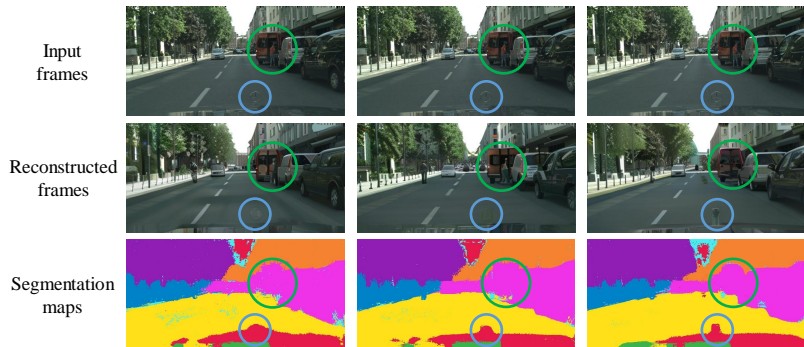

Figure 8: Failure case: inversion. In adjacent frames, the car and the sign of the car experience shape changes, which finally result in obvious shape changes on the segmentation maps. We highlight the main areas of discrepancy in green and blue circles.

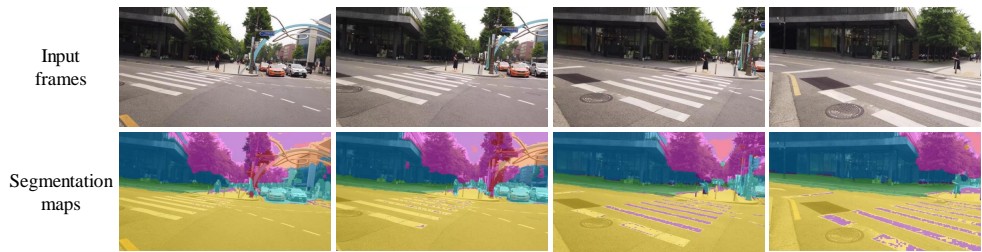

Figure 9: Failure case: temporal inconsistency. The sidewalk is clustered into the yellow region in the first frame, and later the same sidewalk is clustered into another region denoted as purple region.

### A.5 HYPERPARAMETERS

We provide all hyperparameters for SD 2.1 and SVD in Table 3.

### A.6 FAILURE CASES

We identify several typical failure cases using our method. The first one is the inaccurate diffusion inversion process (Figure 8). In adjacent frames, the texture and shape details of the small objects can be different after inversion. Segmentation maps will also inherit this discrepancy between input frames and reconstructed frames, which will result in flickering on the frames. The second one is flickering between similar regions (Figure 9). Some objects are originally assigned with one cluster, while in the later frames, they are grouped into different clusters. The third one is dealing with unseen objects (Fig. 10). As we only use an anchor frame's ground truth to guide the class-agnostic clusterings, a feature-bank based strategy could be adopted to adapt our approach to handle dynamic videos.

### A.7 EFFICIENCY

We report the FPS and GPU memory consumption of our methods on the validation subset of the DAVIS 2017 dataset in Tab. 4. Our approach with SD2.1 as backbone processes 0.41 frames per second, where the primary bottleneck is the multiple forward passes of the diffusion model required for the feature aggregation and modulation steps. It is worth mentioning that SD2.1 and SVD models are not lightweight and not optimized for rapid forward pass compared to vision backbones such as VITs and ResNets. Since our aim was to establish a new paradigm for zero-shot video segmentation based on diffusion models, our main focus was to achieve competitive segmentation accuracy. For real-time applications where achieving the highest efficiency is required, further efforts are needed as a follow-up to our work. These efforts can include training an adapter to perform the

feature aggregation efficiently, similar to Luo et al. (2023a), replacing the modulation process with a feature upsampling module as in Fu et al. (2024), or even fine-tuning Stable Diffusion to function as end-to-end segmentation models.

Table 3: Hyperparameters settings for SD and SVD.

|  | Ours (SD 2.1) | Ours (SVD) |
|---|---|---|
| $t_f$ | 25 | 25 |
| $t_m$ | 20 | 17 |
| Sampling timesteps | 25 | 25 |
| Sampler | EDM | EDM |
| Block $b_k$ | Block 6, 7 and 8 | Block 6, 7 and 8 |
| Block $b_m$ | Block 7 | Block 8 |
| Block $c$ | Block 7 | Block 8 |
| Spatial threshold $\mathcal{T}$ | 1 | 1 |
| Filtering strength $s$ | 0.7 | 0.7 |
| Modulating factor $\lambda$ | 50.0 | 50.0 |
| Modulating attention type | cross attention | self attention |
| Injected features | spatial attention | spatial & temporal attention |

Table 4: Efficiency comparison.

| Method | Backbone | # of parameters | FPS | GPU Memory (GB) |
|---|---|---|---|---|
| EmerDiff | SD2.1 | 865M | 0.44 | 10 |
| Ours | SD2.1 | 865M | 0.41 | 21 |
| Ours | SVD | 1.5B | 0.12 | 39 |

## A.8 Evaluation on Video Object Segmentation dataset

We provide results on the DAVIS 2016 and DAVIS 2017 Video Object Segmentation (VOS) datasets to demonstrate that our method can be applied to other video segmentation tasks. Our approach consistently outperforms EmerDiff by a huge margin on VOS, demonstrating its applicability to other VS tasks.

Table 5: Evaluation on DAVIS 2016 and DAVIS 2017 datasets.

|  | DAVIS 2016 | | | DAVIS 2017 | | |
|---|---|---|---|---|---|---|
|  | $\mathcal{J} \& \mathcal{F}$ | $\mathcal{J}$ | $\mathcal{F}$ | $\mathcal{J} \& \mathcal{F}$ | $\mathcal{J}$ | $\mathcal{F}$ |
| EmerDiff | 22.1 | 20.3 | 23.9 | 18.6 | 15.9 | 21.3 |
| Ours (SVD) | 66.1 | 67.7 | 64.6 | 40.7 | 40.9 | 40.4 |
| Ours (SD2.1) | **79.0** | **70.5** | **69.3** | **60.1** | **57.6** | **62.6** |

## A.9 Long video segmentation

We are not aware of any existing long video semantic segmentation dataset. Therefore, we test our approach on the CLVOS (Nazemi et al., 2023) dataset for Video Object Segmentation (VOS), which has long videos of an average length of 1506 frames. This is significantly longer than the VSPW dataset (71 frames). In theory, there is no restriction on the maximum length of the video, as we process the video frames with a fixed batch size.

We show the results in Tab. 6. Our approach still performs well on these significantly longer videos, while EmerDiff drastically fails. These results highlight the large improvement we made in the zero-shot segmentation setting. Future work can be further improvements on long video segmentation.

Table 6: Evaluation on CLVOS dataset.

|  | $\mathcal{J}$ & $\mathcal{F}$ | $\mathcal{J}$ | $\mathcal{F}$ |
|---|---|---|---|
| EmerDiff | 0.8 | 0.7 | 0.9 |
| Ours (SVD) | 23.2 | 22.1 | 24.3 |
| Ours (SD2.1) | **46.7** | **46.8** | **46.5** |

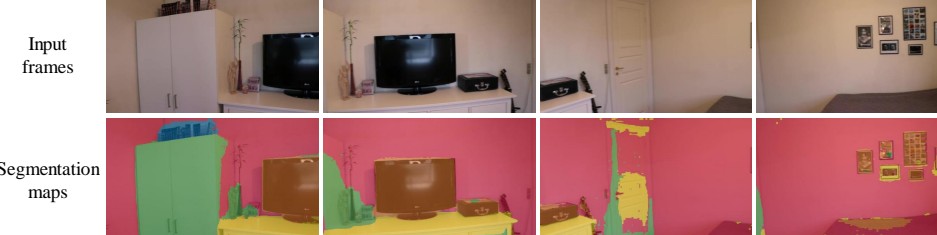

Input frames

Segmentation maps

Figure 10: Failure case: unseen objects. The door and albums, which do not appear in the previous frames, are assigned with the wrong clusters.

## A.10 ABLATION STUDY

We provide additional ablation experiments here. We ablate the modulating Block $b_m$ in Figure 11 for both SD and SVD. Modulating Block 7 and Block 8 for SD and SVD, respectively, can give the most spatial details as well as maintain the semantics. These blocks are, at the same time, the most semantically-riched blocks in SD and SVD. In Figure 12, we show more examples of how latent blending and difference map filtering help with removing spatial noises. In Figure 13, we show comparisons between the difference maps and segmentation maps produced by SD and SVD. We show that the difference maps of SD contain finer details, which bring sharper boundaries to the final segmentation maps. We further show PCA visualization of a $64 \times 64$ resolution block to complement Figure 2 in the main paper. We show that in high-resolution blocks, SD features have more semantic information and spatial details, as well as more temporally stable than SVD spatial features and temporal features. We also show that segmentation maps under different numbers of K-Means clusters and after GT labels reassignment in Figure 14. Increasing the number of clusters can help segment more small objects (from 5 to 20). However, further increasing the number of clusters may not necessarily segment out the clusters that are aligned with defined clusters in ground truth (from 20 to 30). Although K-Means originally generated 30 clusters, most of them are merged to the same classes after GT label reassignment.

We also ablate on the classifier we use in Stage 2 on the first 30 videos of the VSPW validation set in Tab. 7. We opt for low-complexity classifiers for a reduced computational overhead and to avoid overfitting. The results show that both KNN and MLP achieve a good tradeoff between speed and performance. RandomForest performs slightly better but at an increased computational overhead.

Table 7: Ablation on different classifiers

| Classifier | mIoU | mVC$_8$ | mVC$_{16}$ | Speed |
|---|---|---|---|---|
| Adaboost | 34.1 | 82.4 | 79.2 | 1x |
| Random Forest | 47.9 | 90.5 | 89.0 | 148x |
| MLP | 47.1 | 89.1 | 87.6 | 240x |
| KNN | 46.5 | 89.8 | 88.4 | 240x |

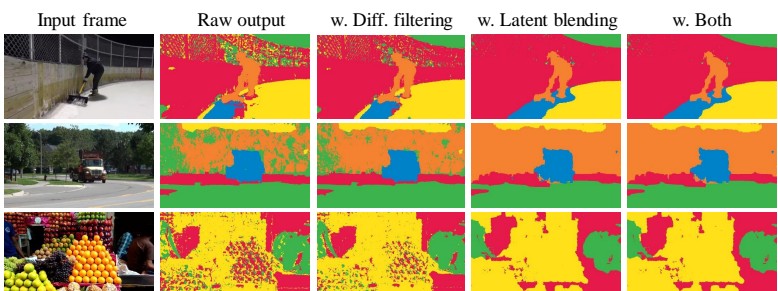

Figure 11: Ablation: modulating different block will result in different segmentation maps. Modulating Block 7 and Block 8 give the best results for SD and SVD, respectively (highlighted in shadow).

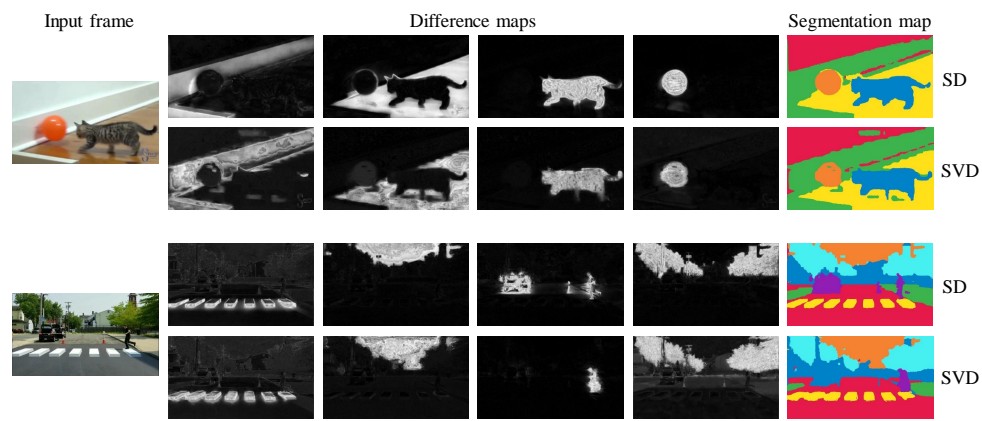

Figure 12: Ablation: latent blending and difference map filtering.

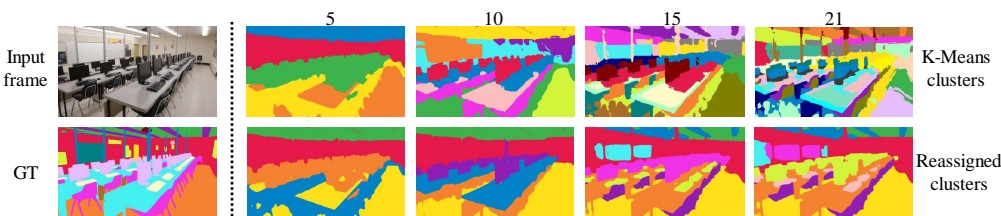

Figure 13: Ablation: Difference maps comparison between SD and SVD.

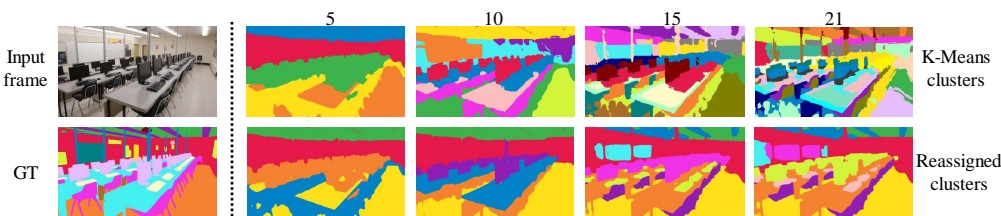

Figure 14: Ablation: number of K-Means clusters. In the first row, we show the segmentation maps generated by different numbers of clusters in K-Means. In the second row, we show the segmentation maps generated by the same numbers of clusters followed by GT labels reassignment.

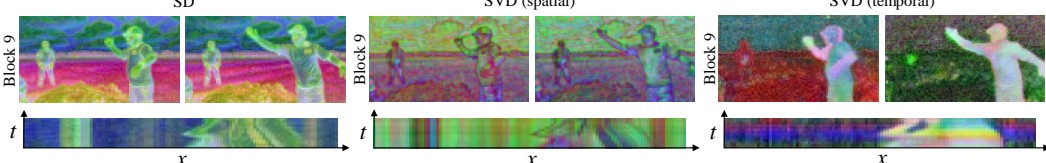

Figure 15: A visualization of the first three PCA components for the features extracted from the most semantically-rich blocks of Block 9 in both SD and SVD of the first and last video frames in a batch. In the second row, we show the $x$-$t$ slice of a set of pixels (highlighted in the red line in the leftmost PCA visualization) horizontally across the PCA visualization ($x$-axis) and stack it chronologically across the full batch of video frames ($t$-axis).

