# OpenReview forum: "Zero-Shot Video Semantic Segmentation based on Pre-Trained Diffusion Models"
_ICLR.cc/2025/Conference — ICLR 2025 Conference Withdrawn Submission_

### Official Review · Reviewer_P8MD · 2024-10-26

**Soundness:** 3
**Presentation:** 3
**Contribution:** 3
**Rating:** 6
**Confidence:** 4

**Summary:**

To address the challenge of existing methods lacking temporal modeling in video data, this paper uses a diffusion model with autoregressive updates to adapt to scene changes, handling the issue of temporal continuity. Its effectiveness has been validated on datasets such as VSPW.

**Strengths:**

1. Using diffusion to model the temporal dimension is a very interesting idea. Predicting a coarse segmentation map for each frame and adapting to scene changes in the video through autoregressive updates is highly intuitive.

2. The writing is good and the proposed method is easy to follow.

3，The experimental results are solid.

**Weaknesses:**

1. The goal of VSS (Video Semantic Segmentation) is to predict the semantic category for each pixel in each frame based on predefined classes. EmerDiff has shown good performance in image segmentation tasks. Would its approach, with the addition of a certain level of temporal modeling, serve as a more suitable baseline or comparison method to validate the effectiveness of the proposed approach?

2. VSS is a highly complex task, and balancing the effectiveness of capturing temporal and spatial information with computational efficiency is key. An important concern is whether the proposed method might increase the computational burden, leading to excessively long training times.

3. Lack comparison and discussion with recent important work on video scene segmentation (VSS) task, General and Task-Oriented Video Segmentation, in ECCV 2024.

4. How was the KNN classifier chosen, and is it the best clustering model?

**Questions:**

1. The inputs and outputs of Latent Blending in the Fig.3 are not very easy to understand. Could you pls explain more?


2. What is the distance metric used in the pixel-level voting mechanism during the cross-frame matching process?

---

### Official Review · Reviewer_tDPN · 2024-10-28

**Soundness:** 2
**Presentation:** 3
**Contribution:** 2
**Rating:** 5
**Confidence:** 3

**Summary:**

This paper introduces a zero-shot video semantic segmentation approach using pre-trained diffusion models. The proposed methods, which involve a CBR strategy, scene context model, and others, enhance spatial and temporal consistency. Experiments demonstrate its good performance on VSS benchmarks.

**Strengths:**

S1: This paper is well written and easy to follow.

S2: This paper explores the characteristics of SD and SVD on video tasks and proposes a novel zero-shot video semantic segmentation method based on Emerdiff. The proposed CBR strategy can effectively enhance the temporal and spatial consistency of coarse segmentation maps.

**Weaknesses:**

W1: The comparisons in experiments are weak and can’t show the superiority of these methods. The baselines that are surpassed are all focused on image segmentation, so it does not effectively demonstrate the method’s strengths. Additionally, although the paper compares its approach with other supervised video segmentation methods, it does not achieve state-of-the-art performance. Furthermore, there are no comparisons with other zero-shot video segmentation methods. Therefore, I suggest adding baselines of other zero-shot video segmentation methods to highlight the advantages of the proposed approach.

W2: I understand one of the main innovations of this paper lies in the CBR strategy, but the ablation study does not provide quantitative results for the CBR ablation and doesn’t have enough analysis for it.

**Questions:**

It is hard to understand why SD features would be more stable than SVD in both spatial and temporal dimensions (as depicted in Appendix A.10). From my perspective, Table 1 shows that EmerDiff (SVD) performs better than EmerDiff (SD) in mVC, suggesting that the feature in SVD should have better temporal consistency. However, SVD underperforms compared to SD in other results, so I agree that, compared to SD, SVD may have a more degraded feature representation (as depicted in Sec 5.3). Meanwhile, can you provide more convincing evidence for this?

---

### Official Review · Reviewer_T9ik · 2024-10-29

**Soundness:** 2
**Presentation:** 3
**Contribution:** 2
**Rating:** 5
**Confidence:** 5

**Summary:**

The proposed paper focus on the problem of zero-shot Video Semantic Segmentation (VSS) based on Diffusion models, Stable Diffusion (SD) and Stable Video Diffusion (SVD) models. Specifically, the authors introduce a zero-shot VSS approach leveraging spatial features from pretrained diffusion models (SD or SVD) learned from large-scale image/video data and a scene context model that adapts segmentation maps over video frames.
This approach includes a correspondence-based refinement (CBR) strategy, based on the semantic correspondence between consecutive frames to enforce better temporal consistency and a masked modulation process to upsample the coarse segmentation masks to the full resolution of the video frames for improved segmentation accuracy. The authors perform experiments on three commonly used VSS datasets: VSPW, Cityscapes and CamVid, to evaluate their proposed zero-shot VSS pipeline.

**Strengths:**

1. The proposed paper is well-written and easy-to-follow. The figure demonstration is very clear.
2. The problem of interest, i.e. (zero-/few- shot) Video Semantic Segmentation is practically meaningful. I agree the authors that VSS is less explored compared to its image (single-frame) counterpart.
3. The proposed paper provides a feasible solution to the proposed problem, given the experimental results shown in Section 5 in the main paper, though certain important aspects are omitted (please refer to the next section for further details).

**Weaknesses:**

1. Technical contributions. The proposed paper seems to heavily rely on existing works, i.e. Emerdiff [Namekata et al., 2024] to extract per-frame semantic features and generate coarse segmentation maps, with additional correspondence-based refinement (CBR) and mask modulation to refine and upsample the coarse segmentation maps. On the other hand, the proposed segmentation map refinement strategy, CBR which employ pixel-wise correspondence has been widely studied in previous work, especially video segmentation related vision tasks, e.g. Tang et al. (2023) as the authors pointed out in the paper and [a-c] etc. While the mask modulation operation is essentially a weighted summation operation to mask out latents and difference map, the overall technical contributions c.f. video semantic segmentation seems incremental and lacking significance.
2. Sufficiency of Experiments. While the focus of the proposed work is zero-shot VSS, several aspects are missing to convincingly support authors' claims. a) It is important to show the performance gain of the proposed pipeline versus naive frame feature aggregation to prove the effectiveness of the proposed CBR and mask modulation. I.e., what is the performance like with SD spatial features + average/max pooling of the features for the final video-level segmentation? Does row 1 and 2 in Table 2 refer to a similar case? If so, it would be better to be clarified in the paper. b) While "zero-shot" being one of the major contributions claimed in the paper, it would be important to compare the proposed pipeline with other foundation models like SAM/SAM2. One additional minor point is that, the proposed pipeline requires GT labels to assign class labels to the segmentation map for the first frame, whether the proposed pipeline is fully "zero-shot" may be argueable.
3. Computation Efficiency. The proposed pipeline for processing videos requires to process per-frame features, while also computing pixel-level correspondence (in the proposed CBR module), which is quite computationally intensive. This has also validated by the authors in the supplementary material. There has been works explored object-level correspondence for more efficient computation, e.g. [b], I wonder if the authors can provide more insights from this aspect.
4. Minor: many references are missing the conference information.

[a] Cheng et al., Rethinking Space-Time Networks with Improved Memory Coverage for Efficient Video Object Segmentation, NeurIPS 2021.
[b] Cheng et al., Putting the Object Back into Video Object Segmentation, CVPR 2024.
[c] Sun et al., Alignment before aggregation: trajectory memory retrieval network for video object segmentation, ICCV 2023.

**Questions:**

Please refer to "Weaknesses" section for further details.

**Details Of Ethics Concerns:**

I don't see any concerns here.

---

### Official Review · Reviewer_Wyv4 · 2024-11-02

**Soundness:** 2
**Presentation:** 2
**Contribution:** 4
**Rating:** 5
**Confidence:** 4

**Summary:**

Authors propose a technique for unsupervised video semantic segmentation that is relying on diffusion models. They propose a context model that relies on a KNN classifier and a correspondence-based refinement module that that ensures the temporal consistency of the labels. The method is evaluated on VSPW benchmark and compared to single image models that mine diffusion models for segmentation.

**Strengths:**

- The novelty of their contribution as the first to mine diffusion models for video semantic segmentation
- Consistent strong gains on three acknowledged benchmarks Cityscapes, CamVid and VSPW.

**Weaknesses:**

- There is an inconsistent behaviour in the results that needs to be explained. Their results show better performance on VSPW than Cityscapes, when it is widely acknowledged that VSPW is much harder dataset with more categories. For example refer to OCRNet[1] results on VSPW at 36.68% while on Cityscapes at 81.8%. While these are fully supervised models it is still not explained why unsupervised ones will act differently? It seems something is wrong in the setup. Even if the claim this has to do with the dataset resolution, that wouldn't be the case on CamVid and still why fully supervised methods face more challenges in VSPW than Camvid/Cityscapes.

- Is there a way to setup their baseline for EmerDiff to just be evaluated per image? is there a reason why they are only reporting the adaptation to the video segmentation with their proposed context model. Currently, it seems their baseline is still dependant on their own adaptation so a simple comparison to image based baselines that rely on diffusion models will serve the purpose of confirming the benefit from their video based approach.

- It would be interesting to see its performance w.r.t DiffSeg[2] another image based segmentation mining diffusion models with focus on attention maps.


- While not sure but seems their ablation is conducted on part of the VSPW val set, why 30 videos only? os is this the full number of videos in the val set?

[1] Yuan, Yuhui, et al. "Segmentation Transformer: Object-Contextual Representations for Semantic Segmentation. 2021." 1909.

[2] Frick, Raphael Antonius, and Martin Steinebach. "DiffSeg: Towards Detecting Diffusion-Based Inpainting Attacks Using Multi-Feature Segmentation." Proceedings of the IEEE/CVF Conference on Computer Vision and Pattern Recognition. 2024.

**Questions:**

Questions were noted in the above weaknesses.

- Also it is not clear how the context model operates, how KNN is trained exactly? It is a nonparametric technique so it is not clear how in their figure they refer to it as undergoing some fine-tuning or adaptation when there are no parameters. An explanation of how this is done would be more suitable for that.

- In their evaluation protocol they do use the labels in the first frame to assign labels, so is this really considered zero shot? Are these labels used during the correspondence refinement. Or this is solely some form of Hungarian matching during evaluation only? Why not just do this Hungarian matching per image why only first frame? I am wondering if something is wrong in the evaluation protocol that lead to the inconsistency above between VSPW vs. CamVid.

---

### Official Review · Reviewer_88Fd · 2024-11-04

**Soundness:** 2
**Presentation:** 3
**Contribution:** 1
**Rating:** 3
**Confidence:** 3

**Summary:**

This paper introduces a zero-shot video semantic segmentation (VSS) method based on diffusion models. The method consists of 3 main components: 1. a scene context model that models the classes in a video based on diffusion model decoder features 2. a correspondence algorithm that produces a coarse segmentation map for each frame and 3. an upsampling strategy that applies masking to modulated segmentation masks. The method is implemented using either Stable Diffusion and Stable Video Diffusion as backbones. Experiments are conducted to compare the method's performance to the baselines on 3 VSS datasets. Ablations are carried out to asses the contribution of the 3 components above on a subset of a VSS dataset.

**Strengths:**

1. The method's contribution in terms of technical novelty is strong: the masked modulation strategy, correspondence algorithm and scene context model are novel.
2. The method is well-presented so that the method's components are clear. For the most part the paper is clearly written.

**Weaknesses:**

1. The label setting and motivation are not clear. In Section 5.1, Evaluation Protocol, it says that the labels from the ground truth of the first frame are used to assign labels to the method's clusters (l. 359-360). However, the paper claims that the method is zero-shot [1], leading to confusion on the task the method is designed for. If the method is using the first frame of the video to assign classes then it should be categorized as Video Object Segmentation [2] and compared to the baselines for this task, rather than Video Semantic Segmentation.
2. The method is not well-validated and deserves further analysis. In Table 2, it is not clear based on the difference in mIOU and mVC whether all components are necessary to achieve the best performance. The analysis does not delve into the contributions of different components (e.g. when the feature aggregation can fail: in l. 472 it says "some occasions" but what occasions?) Other aspects of the model were not validated, such as choosing the features from blocks 6-8.
3. The computational efficiency (training time, inference time and memory use on the GPU) of the non-diffusion-based baseline methods are missing. This is important in assessing the ease-of-use of the method and contributes to one's decision to apply the method.

[1] "Zero-Shot Semantic Segmentation". Maxime Bucher, Tuan-Hung Vu, Matthieu Cord and Patrick Pérez. NeurIPS, 2019.
[2] "The 2017 DAVIS Challenge on Video Object Segmentation". Jordi Pont-Tuset, Federico Perazzi, Sergi Caelles, Pablo Arbeláez, Alex Sorkine-Hornung and Luc Van Gool. Arxiv:1704.00675, 2018.

**Questions:**

Questions:
1. In l. 107 what evidence is there that training dataset diversity and size will affect the trend that SD features leads to higher quality results than SVD features?
2. Is L in l. 177 set to the number of classes in the dataset? If so how does the model deal with segmenting separate instances of the same class in the same mask?
3. The exposition of the scene context model would benefit greatly from a set of equations showing clearly what the input and output of $\Omega$ is.
4. Why was cosine similarity chosen in l. 267 and not L1 or L2?
5. Can you give an intuition on the two cases in parametric Equation 4?
5. What is $B$ in Equation 5? If it's the number of batches why does the sum exclude the last batch?
6. In l. 429-430 why doesn't the masked modulated upsampling address this?

Minor typos/Writing Suggestions:
- Define what modulation is in the beginning of the paper.
- l. 166 "2024), We"-> "2024), we"
- l. 167 "where We"-> "where we"
- l. 267 "unplausible"->"implausible"
- Figure 4 caption would benefit from a more in depth description of the scene context model.
- l. 428 "their"->"the"
- l. 458 "produce"->"produces"

---

### Note · Authors · 2024-11-15

I have read and agree with the venue's withdrawal policy on behalf of myself and my co-authors.